elLIFE

# Small molecule Photoregulin3 prevents retinal degeneration in the $Rho^{P23H}$ mouse model of retinitis pigmentosa

**Paul A Nakamura[1], Andy A Shimchuk[1], Shibing Tang[2], Zhizhi Wang[1], Kole DeGolier[1], Sheng Ding[2], Thomas A Reh[1]***

[1]Department of Biological Structure, University of Washington, School of Medicine, Seattle, United States; [2]Department of Pharmaceutical Chemistry, UCSF School of Pharmacy, University of California, San Francisco, San Francisco, United States

**Abstract** Regulation of rod gene expression has emerged as a potential therapeutic strategy to treat retinal degenerative diseases like retinitis pigmentosa (RP). We previously reported on a small molecule modulator of the rod transcription factor Nr2e3, Photoregulin1 (PR1), that regulates the expression of photoreceptor-specific genes. Although PR1 slows the progression of retinal degeneration in models of RP in vitro, in vivo analyses were not possible with PR1. We now report a structurally unrelated compound, Photoregulin3 (PR3) that also inhibits rod photoreceptor gene expression, potentially though Nr2e3 modulation. To determine the effectiveness of PR3 as a potential therapy for RP, we treated $Rho^{P23H}$ mice with PR3 and assessed retinal structure and function. PR3-treated $Rho^{P23H}$ mice showed significant structural and functional photoreceptor rescue compared with vehicle-treated littermate control mice. These results provide further support that pharmacological modulation of rod gene expression provides a potential strategy for the treatment of RP.
DOI: https://doi.org/10.7554/eLife.30577.001

*For correspondence:
tomreh@uw.edu

## Introduction

Retinitis pigmentosa (RP) is an inherited retinal degenerative disease with a prevalence of 1 to 3,000-5,000 births (*Hartong et al., 2006*; *Parmeggiani, 2011*; *Boughman et al., 1980*). More than 3,000 mutations in about 60 genes have been identified to be associated with RP (*Hartong et al., 2006*; *Daiger et al., 2013*). Most of these mutations are in genes essential for rod photoreceptor development and function (*Hartong et al., 2006*). There is currently no approved medical therapy that slows or prevents rod degeneration in these individuals.

One emerging approach to treating retinal degeneration is through targeting the factors that regulate rod gene expression. Studies of retinal development have identified several transcription factors that regulate photoreceptor gene expression. For example, loss of function mutations in the rod-specific transcription factors Nrl or Nr2e3 cause rods to acquire a more cone-like identity (*Mears et al., 2001*; *Montana et al., 2013*; *Haider et al., 2006*; *Haider et al., 2000*; *Cheng et al., 2011*; *Cheng et al., 2004*; *Corbo and Cepko, 2005*; *Peng et al., 2005*; *Chen et al., 2005*; *Zhu et al., 2017*; *Yu et al., 2017*). Knockout/down strategies have shown that Nrl and Nr2e3 are necessary even in mature rods to maintain their normal levels of gene expression (*Montana et al., 2013*; *Zhu et al., 2017*; *Yu et al., 2017*). Moreover, the reductions in rod gene expression from deletion of *Nrl* or *Nr2e3*, with either conditional deletion or CRISPR-Cas9 deletion, were sufficient to promote the survival of photoreceptors in multiple models of recessive and dominant RP (*Montana et al., 2013*; *Zhu et al., 2017*; *Yu et al., 2017*).

**eLife digest** There are several diseases that cause people to lose their eyesight and become blind. One of these diseases, called retinitis pigmentosa, kills cells at the back of the eye known as rod cells. At first, it affects vision in low light and peripheral vision, but later it affects vision during the daytime as well. There are no effective treatments for patients with retinitis pigmentosa. Yet previous genetic studies have shown that disrupting the activity of genes in rod cells can slow the progression of the disease and preserve vision in mice.

As for all genes, proteins called transcription factors regulate the activity of rod cell genes. Nakamura et al. now report the discovery of a small drug-like molecule, that they name Photoregulin3, which alters the activity of a transcription factor that regulates rod genes. In follow-up experiments, mice with a mutation that replicates many of the features of retinitis pigmentosa were given Photoregulin3 to see if it could slow the progression of the disease. Indeed, Photoregulin3 could stop many of the rod cells from degenerating in the treated mice. At the end of the experiment, the mice treated with this small molecule had about twice as many rods as the control mice. The treated mice also responded better to flashes of light.

Nakamura et al. hope that the findings will one day benefit patients with retinitis pigmentosa. But first more research needs to be done before testing Photoregulin3 in humans. For example, the drug-like molecule needs to be made more potent, and if possible adapted to work when given orally, meaning patients could take it as a pill.

DOI: https://doi.org/10.7554/eLife.30577.002

We have recently reported that this regulatory pathway can be modulated using small molecule modulators of rod gene expression that we have named Photoregulins (*Nakamura et al., 2016*). Treatment of developing or mature retina with Photoregulin1 (PR1) reduces rod gene expression and increases the expression of some cone genes. In addition, treatment of two mouse models of RP (mice with the $Rho^{P23H}$ and the $Pde6b^{Rd1}$ mutations) with PR1 slows rod degeneration in vitro (*Nakamura et al., 2016*). However, in vivo analyses of PR1 were limited by the compound's potency, solubility, and stability in vivo. In this study, we have identified a structurally unrelated compound, Photoregulin3 (PR3) that also significantly represses rod gene expression, but is more amenable for in vivo studies. With PR3 treatment, we show anatomical and functional preservation of the retina in $Rho^{P23H}$ mice, providing in vivo proof-of-concept of this novel therapeutic strategy for the treatment of RP.

## Results and discussion

### In vitro characterization of PR3

In order to identify compounds that may target Nr2e3, we searched PubChem (https://pubchem.ncbi.nlm.nih.gov/) for top-scoring hit compounds previously identified to interact with Nr2e3 in transfected CHO-S cells in a luciferase-based assay (PubChem Assay IDs: 602229, 624378, 624394, and 651849). As a secondary screen for the initial hits we used primary retinal cell cultures and assayed Rhodopsin expression because it is a well-defined target of Nr2e3 signaling and is expressed at high levels exclusively in rod photoreceptors (*Cheng et al., 2004*; *Peng et al., 2005*; *Haider et al., 2009*). We dissociated retina from postnatal day 5 (P5) mice and cultured them in media containing the small molecules. After treatment for 2 days, we assessed Rhodopsin expression with an immunofluorescence-based assay (*Nakamura et al., 2016*). One compound, PR3 (*Figure 1A*), showed robust reduction in Rhodopsin compared to DMSO and PR1 treatment (*Figure 1B*). We confirmed this finding with qPCR analysis using intact retinal explant cultures from P4 mice treated with a 0.3 μM dose of PR1 or PR3. Similar to the immunofluorescence assay, treatment with PR3 resulted in reduced *Rhodopsin* but not *Otx2*, a rod transcription factor upstream of Nr2e3, compared to DMSO and PR1 (*Figure 1C*).

Mutations in *Nr2e3* result in an increased number of S Opsin+ photoreceptors as well as a reduction in rod gene expression (*Haider et al., 2006*; *Haider et al., 2000*; *Cheng et al., 2011*; *Peng et al., 2005*; *Chen et al., 2005*; *Cheng et al., 2006*). To determine if PR3 treatment also

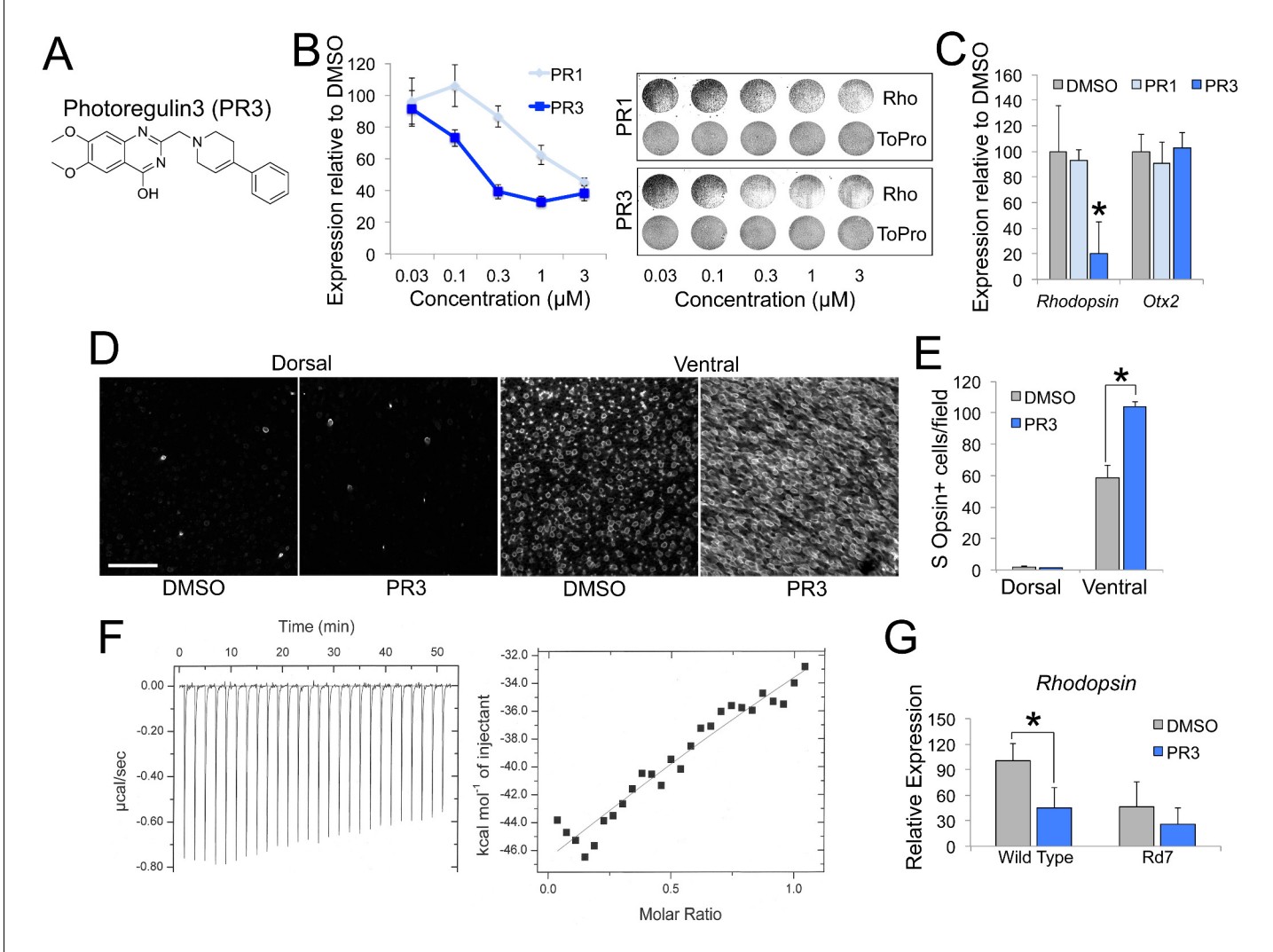

**Figure 1.** PR3 reduces rod gene expression via Nr2e3. (**A**) Chemical structure of Photoregulin3 (PR3). (**B**) Dose-response relationship of PR1 and PR3 on Rhodopsin expression in dissociated retinal cell cultures (n = 3 for each concentration for each compound; value graphed is the mean ±SEM of 3 biological replicates). Example scans of Rhodopsin and ToPro3 staining from retinal cell cultures are shown on the right. (**C**) Confirmation of PR3's potency by qPCR from intact retinal explant cultures from P4 mice treated with DMSO or 0.3 μM PR1 or PR3 for 2 days (n = 3–4 for each compound; value graphed is the mean ±SEM of the biological replicates, *p=0.0075 for DMSO vs. PR3, two-tailed *t*-test assuming equal variance using $\Delta C_T$ values). (**D**) Intact retinas from P11 mice were explanted in media containing DMSO or 0.3 μM PR3 for 3 DIV and then stained for S Opsin in a whole-mount preparation. Scale bar represents 50 μm. (**E**) PR3-treated retinas had more S Opsin+ cells per 100 μm x 100 μm field in the ventral, but not dorsal, retina compared to DMSO-treated retinas (n = 3 biological replicates; value graphed is the mean ±SEM of the biological replicates, *p=0.00063, two-tailed *t*-test assuming equal variance). (**F**) Isothermal titration calorimetric study of PR3 binding to Nr2e3. (**G**) Intact retinas from adult (P23–P35) wild type and Rd7 were explanted in media containing DMSO or 1 μM PR3 for 2 days and *Rhodopsin* expression was measured by qPCR. PR3 decreased *Rhodopsin* in wild type retinas but not in Rd7 retinas (n = 4 biological replicates; value graphed is the mean ±SEM of the biological replicates, *p=0.033, two-tailed *t*-test assuming equal variance for wild type and p=0.13, two-tailed *t*-test assuming equal variance for Rd7 using $\Delta C_T$ values).

DOI: https://doi.org/10.7554/eLife.30577.003

The following source data and figure supplements are available for figure 1:

**Source data 1.** Source data for *Figure 1*.
DOI: https://doi.org/10.7554/eLife.30577.006
**Figure supplement 1.** Effects of PR3 on photoreceptor promoter activation and co-activator binding.
DOI: https://doi.org/10.7554/eLife.30577.004
**Figure supplement 1—source data 1.** Source data for *Figure 1—figure supplement 1*.
DOI: https://doi.org/10.7554/eLife.30577.005

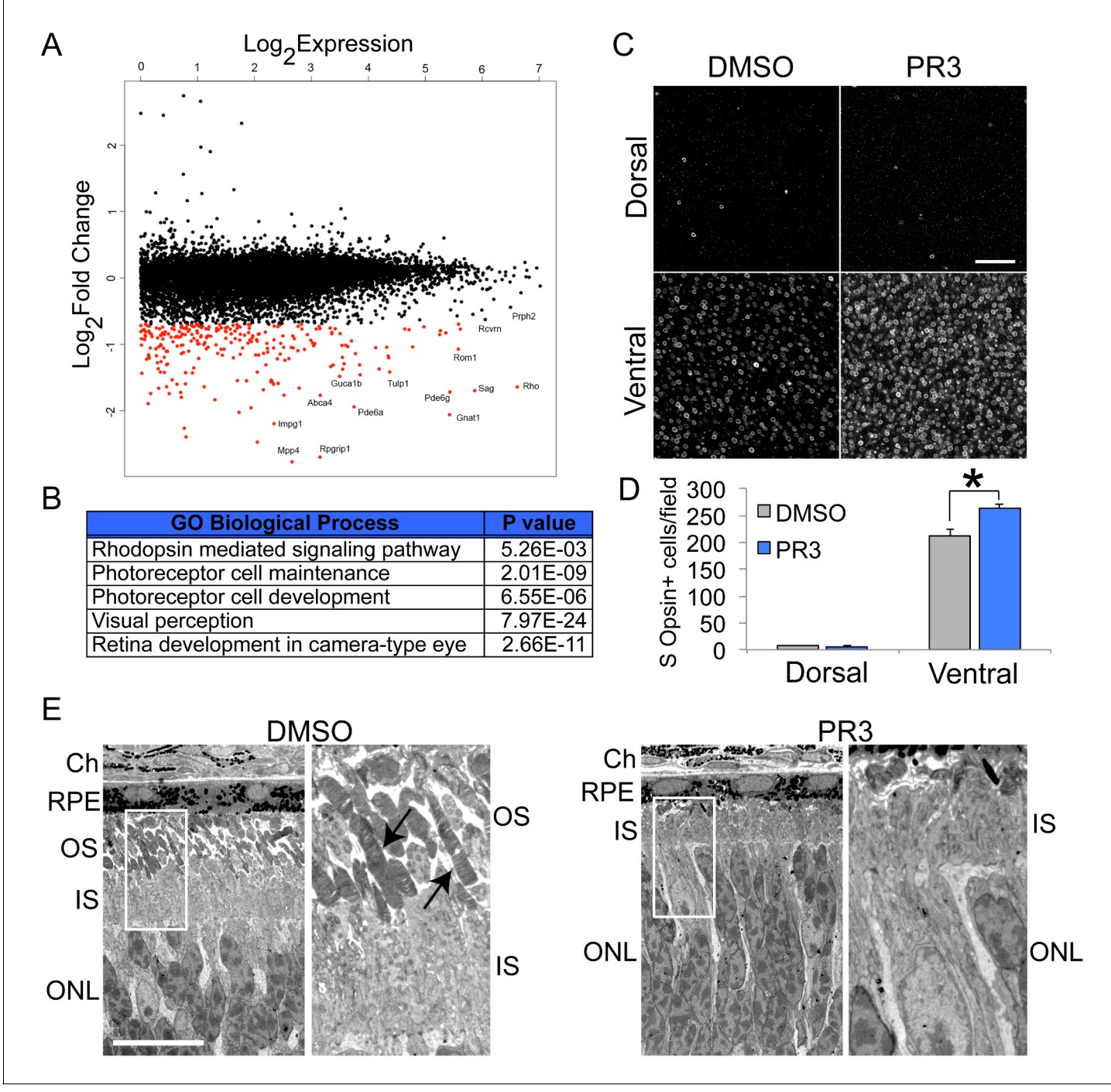

**Figure 2.** In vivo characterization of PR3 treatment. (**A**) RNA sequencing results plotting log2FoldChange against log2Expression (RPKM) of wild type mice treated with DMSO vehicle or 10 mg/kg PR3 shows robust reduction in rod photoreceptor genes. (n = 2 mice per condition; values graphed are the mean of the two biological replicates) (**B**) Gene ontology analysis (http://geneontology.org/page/go-enrichment-analysis) results for largest changes (top 100) in gene expression assessed by RNA sequencing. (**C**) Whole mount S Opsin staining of retinas from mice treated with 10 mg/kg PR3 or DMSO vehicle for 3 days. Scale bar represents 25 µm. (**D**) Retinas from mice treated with PR3 had more S Opsin+ cells per 100 µm x 100 µm field in the ventral, but not dorsal, retina compared to controls (n = 4 biological replicates; value graphed is the mean ±SEM of the biological replicates, *p=0.0086, two-tailed t-test assuming equal variance). (**E**) Electron microscope micrographs of retinal sections of wild type mice treated with DMSO or 10 mg/kg PR3 (n = 2 mice per condition). Compared to DMSO controls, PR3 retinas have arrested outer segment development (arrows indicate outer segments in DMSO treated mice). Scale bare represents 10 µm.

DOI: https://doi.org/10.7554/eLife.30577.007

*Figure 2 continued on next page*

*Figure 2 continued*

The following source data is available for figure 2:

**Source data 1.** Source data for *Figure 2*.

DOI: https://doi.org/10.7554/eLife.30577.008

affects cone gene expression, we explanted intact retinas from P11 wild type mice in media containing DMSO or PR3 for 3 days. We used intact retinas for this experiment to assess changes in dorsal and ventral retina independently. After fixation and whole mount immunostaining, we counted S Opsin+ cells in the dorsal and ventral retina. Similar to *Nr2e3* mutations, treatment with PR3 resulted in an increase in the number of S Opsin+ cells in the ventral, but not dorsal retina (*Figure 1D–E*).

PR3 was initially identified as a chemical modulator of Nr2e3 in a luciferase-based assay that identified ligands by disruption of the Nr2e3-NCoR dimer complex and had a calculated $IC_{50}$ of 0.07 μM in this assay (PubChem Assay IDs: 602229, 624378, 624394, and 651849). To confirm a direct Nr2e3-PR3 interaction, we used isothermal titration calorimetry (ITC). Consistent with our other assays, ITC qualitatively showed a direct interaction between PR3 and Nr2e3 (*Figure 1F*; estimated $K_d$ of 67 μM using a one site model). The initial screen that identified PR3 demonstrated its effects on Nr2e3 in a co-repression assay with NCoR; however, the effects we observed on rod gene expression suggested that PR3 also inhibits the co-activator function of Nr2e3. To explore this possibility, we assessed the effects of PR3 on the ability of Nr2e3 to cooperate with Nrl and Crx, two other

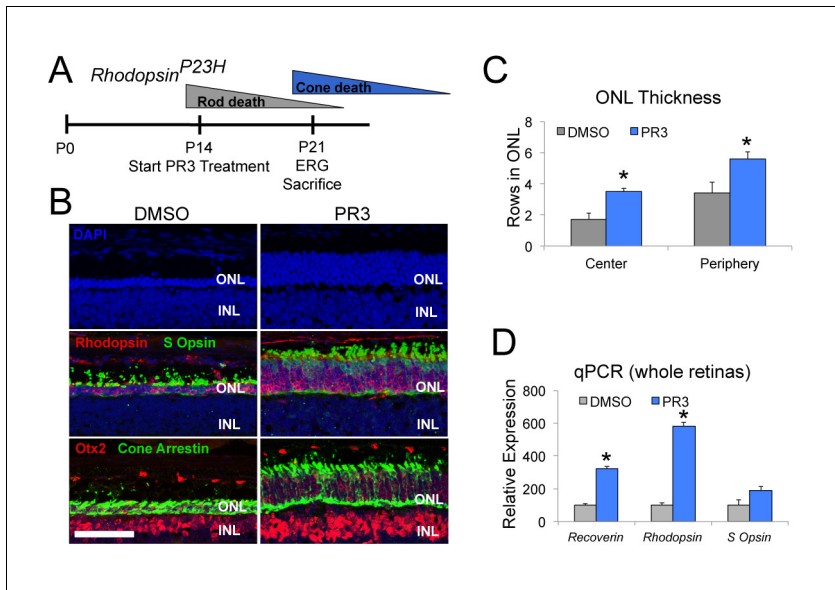

**Figure 3.** PR3 slows the progression of photoreceptor degeneration in the RhoP23H mouse model of RP. (**A**) Timeline of photoreceptor degeneration in the *Rhodopsin^P23H* mouse and experimental design. (**B**) Immunofluorescence staining for Rhodopsin, S Opsin, Otx2, and Cone Arrestin on retinal sections from *Rhodopsin^P23H* mice demonstrate preservation of photoreceptors with PR3 treatment. Scale bar represents 50 μm. (**C**) Counts for rows of DAPI+ cells in the central and peripheral ONL show greater survival of photoreceptors with PR3 treatment (n = 7 mice for DMSO treatment and 8 mice for PR3; values graphed are the mean ±SEM of the biological replicates, p=0.0014 for center and 0.015 for periphery, two-tailed *t*-test assuming equal variance). (**D**) qPCR on whole retinas from *Rhodopsin^P23H* mice treated with DMSO or PR3 shows greater expression of photoreceptor genes *Recoverin*, *Rhodopsin*, and *S Opsin* with PR3 treatment (n = 6 mice for DMSO and 7 mice for PR3; values graphed are the mean ±SEM of the biological replicates, p=1.73E-05 for *Recoverin*, 3.13E-05 for *Rhodopsin*, and 0.076 for *S Opsin*, two-tailed *t*-test assuming equal variance on $\Delta C_T$ values).

DOI: https://doi.org/10.7554/eLife.30577.009

The following source data is available for figure 3:

**Source data 1.** Source data for *Figure 3*.

DOI: https://doi.org/10.7554/eLife.30577.010

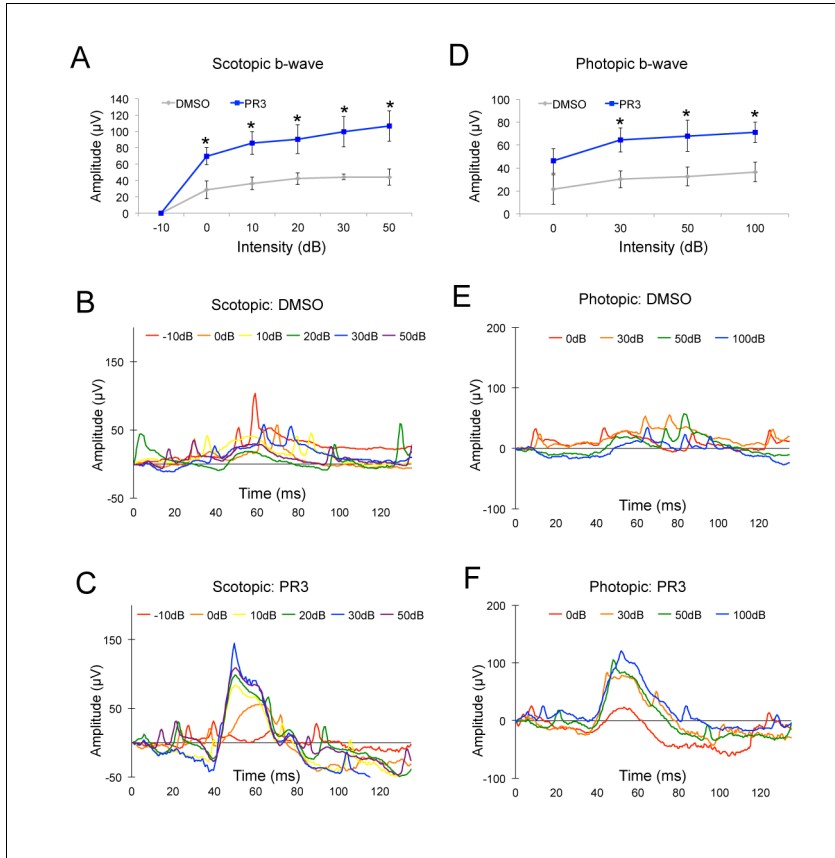

**Figure 4.** PR3 preserves visual function in the RhodopsinP23H mouse. (**A**) Scotopic b-wave amplitudes from P21 *Rhodopsin*[P23H] mice treated with 10 mg/kg PR3 or DMSO vehicle (n = 4 mice for DMSO treatment and 8 mice for PR3 treatment, values graphed are the mean ±SEM of the biological replicates, p=0.025, 0.012, 0.032, 0.021, and 0.015 for 0, 10, 20, 30, and 50 Decibels (dB) respectively, two-tailed *t*-test assuming unequal variance). (**B**) Representative scotopic ERG waveforms from a single DMSO vehicle mouse. (**C**) Representative scotopic ERG waveforms from a single PR3–treated mouse. (**D**) Photopic b-wave amplitudes from P21 *Rhodopsin*[P23H] mice treated with 10 mg/kg PR3 or DMSO vehicle (n = 4 mice for DMSO treatment and 8 mice for PR3 treatment, values graphed are the mean ±SEM of the biological replicates, p=0.18, 0.025, 0.048, and 0.021 for 0, 30, 50, and 100 dB respectively, two-tailed *t*-test assuming unequal variance). (**E**) Representative photopic ERG waveforms from a single control mouse. (**F**) Representative photopic ERG waveforms from a single PR3-treated mouse.
DOI: https://doi.org/10.7554/eLife.30577.011

The following source data is available for figure 4:

**Source data 1.** Source data for *Figure 4*.
DOI: https://doi.org/10.7554/eLife.30577.012

transcription factors known to form a co-activation complex for rod gene expression. We first trans-fected HEK cells with Nr2e3, Nrl, and Crx and measured activation of the *Rhodopsin* promoter in the presence of DMSO or PR3. PR3 strongly reduced *Rhodopsin* promoter activation (*Figure 1—figure supplement 1A*). We next used co-immunoprecipitation to assess whether PR3 affects the inter-action between Nr2e3 and Nrl in HEK 293 cells. We found that PR3 caused an increase in the binding of Nr2e3 to Nrl (*Figure 1—figure supplement 1B–C*). Several reports have shown that ligand binding to nuclear receptors stabilizes interactions with co-activators by stabilizing the struc-ture of the ligand-binding domain and co-activator binding sites (*Onishi et al., 2010*; *Jia et al., 2009*; *Forrest and Swaroop, 2012*; *Fu et al., 2014*). It is not clear why stabilizing the complex would reduce its activity; however, it may be that stabilizing interactions with Nr2e3 prevents Nrl and Crx from interacting with consensus sites on the DNA, or alternatively prevents the recruitment of other components of the transcriptional machinery. Nevertheless, our results indicate that PR3 affects the formation or stabilization of the complex among these critical rod gene regulators.

Although PR3 was identified in an assay for Nr2e3, it is also possible that it interacts with other nuclear receptors. Two related nuclear receptors that are expressed in photoreceptors are *Errb* and *Rorb*. Deletion of these genes in mice has effects on rod gene expression (*Onishi et al., 2010*; *Jia et al., 2009*; *Forrest and Swaroop, 2012*; *Fu et al., 2014*). However, small molecule modulators of ERRb induce rapid rod death (*Onishi et al., 2010*), and so it is unlikely that PR3 acts through this pathway. Loss of function *Rorb* mutations reduce rod gene expression and lead to a phenotype more like what we observe with PR3 treatment. To determine whether PR3 might also act through antagonism of RORb we transfected HEK 293 cells with RORb and Crx and measured activation of the *S Opsin* promoter, a known target of RORb. In transfected HEK 293 cells, RORb and Crx synergistically activate the *S Opsin* promoter (*Srinivas et al., 2006*; *Liu et al., 2017*). Treatment with PR3 did have a small effect on RORb-Crx driven activation of the *S Opsin* promoter (*Figure 1—figure supplement 1D*); however, this effect was much smaller than what we found for PR3 on activation of the *Rhodopsin* promoter by Nr2e3, Nrl, and Crx (*Figure 1—figure supplement 1A*). To further evaluate a role for PR3 as an RORb antagonist, we analyzed downstream RORb target gene expression in developing retina. One target that is specific to RORb, but not regulated by Nr2e3 is *Prdm1* (*Liu et al., 2017*; *Wang et al., 2014*; *Mills et al., 2017*). *Prdm1* expression was unchanged in P0 retinal explants treated with PR3 (*Figure 1—figure supplement 1E*), supporting our hypothesis that the effects of PR3 are primarily mediated through modulation of Nr2e3.

To further evaluate whether PR3 has other targets in the retina, we tested PR3 in Rd7 retinas, which harbor a spontaneous mutation in *Nr2e3* (*Akhmedov et al., 2000*; *Haider et al., 2001*; *Chen et al., 2006*). We explanted adult (P23-35) retinas from wild type and Rd7 mice in media containing DMSO or PR3 and measured *Rhodopsin* expression by qPCR. Consistent with the hypothesis that the effects of PR3 on rod gene expression are mediated through Nr2e3, PR3 caused a significant reduction in *Rhodopsin* expression in wild type retinas but not in retinas from Rd7 mice (*Figure 1G*). While this experiment confirms Nr2e3 specificity of PR3, it remains unclear why Rd7 retinas exhibit only moderate reductions in rod gene expression (*Corbo and Cepko, 2005*; *Peng et al., 2005*; *Chen et al., 2005*; *Haider et al., 2001*), while PR3 treatment or CRISPR-Cas9 deletion of *Nr2e3* (*Zhu et al., 2017*) result in substantial decreases. This suggests that there is some developmental compensation that is not present when the gene is deleted or inhibited in postmitotic rods; alternatively, PR3 may be acting in a dominant-negative manner, and inhibiting the ability of the Nr2e3-Nrl-Crx complex to function properly.

## In vivo characterization of PR3

Nr2e3 signaling is important for rod photoreceptor cell fate, development and maturation, and maintenance of expression. To determine the effect of PR3 treatment on gene expression in postmitotic retinal cells, we systemically treated (intraperitoneal injection; IP) wild type mice with PR3 or vehicle at P12. At P13, 24 hr after the injection, we collected the retinas for global transcriptome analysis by RNA sequencing. As hypothesized, we observed a decrease in most rod photoreceptor-specific transcripts (*Figure 2A–B*). Similar to conditional knockout of *Nrl* in adult mice and knockdown of *Nrl* by CRISPR/Cas9 in postmitotic photoreceptors, we did not observe global increases in cone gene expression (*Montana et al., 2013*; *Yu et al., 2017*). Nevertheless, we did observe an increase in the number of S Opsin+ cells in the ventral retina with PR3 treatment in vivo (*Figure 2C–D*). Genes expressed in both rod and cone photoreceptors, eg. *Crx* and *Otx2*, showed no difference in expression between control and PR3 treatment.

We next used electron microscopy (EM) to perform an ultrastructural analysis of photoreceptor morphology after PR3 treatment. We carried out IP injections of vehicle or PR3 in wild type mice for three consecutive days starting at P11. At P14, 24 hr after the third injection, we euthanized the mice and processed their retinas for EM. Photoreceptor outer segments begin to form during the second and third postnatal week; mutations in *Nr2e3* lead to an impairment in rod outer segment formation, and we predicted that PR3 treatment would affect their development in a similar way (*Haider et al., 2006*). As predicted, PR3 treatment prevented outer segment development; outer segments of PR3-treated photoreceptors were strikingly truncated compared to controls (*Figure 2E*). We did not observe any indication of photoreceptor apoptosis induced by PR3 treatment upon examination of outer nuclear layer (ONL) nuclei (*Figure 2E*), further indicating that the effects on rod development were not due to an increase in cell death.

We have recently shown that reductions in rod gene expression caused by treatment with PR1 were sufficient to slow the degeneration of $Rho^{P23H}$ photoreceptors in vitro (*Nakamura et al., 2016*). The $Rho^{P23H}$ mutation causes misfolding of Rhodopsin in rod photoreceptors, which leads to activation of the unfolded protein response and eventually results in rod and cone death (*Nguyen et al., 2014*). In $Rho^{P23H}$ mice, most rod photoreceptors undergo apoptosis by the end of the third postnatal week (*Sakami et al., 2011*).

To determine whether we could prevent photoreceptor degeneration in this model, we treated $Rho^{P23H}$ mice with PR3 or vehicle from P12-P14 until P21, during the period of rod photoreceptor death (*Figure 3A*). At P21, we assessed visual function with ERGs and euthanized the mice for histological and qPCR analyses. At P21, control $Rho^{P23H}$ mice had only 2–3 rows of photoreceptors remaining in their ONL (*Figure 3B–C*). Rods were sparse and there were few remaining cones (S Opsin+ and Cone Arrestin+ photoreceptors). Control $Rho^{P23H}$ mice had minimal scotopic and photopic b-wave amplitudes by ERG analysis (*Figure 4A, B, D and E*). By contrast, retinas from PR3-treated $Rho^{P23H}$ mice had several rows of rod and cone photoreceptors in the ONL (*Figure 3B–C*). The surviving cones in the PR3-treated retinas were more elongated and healthier than in the DMSO control retinas. We confirmed our histological results with qPCR on retinas from control and PR3 $Rho^{P23H}$ mice and found that treated mice had more expression of *Recoverin* and *Rhodopsin*, indicating greater photoreceptor cell survival (*Figure 3D*). ERG analysis of PR3-treated $Rho^{P23H}$ mice showed significantly elevated scotopic and photopic b-wave amplitudes at most stimulation intensities compared to littermate controls (*Figure 4A, C, D and F*). Together, these data support the conclusion that PR3 treatment prevented structural and functional degeneration of photoreceptors in this model of RP.

In this study we successfully prevented photoreceptor degeneration in the $Rho^{P23H}$ mouse, the first report of successful treatment of this RP model with a small molecule in vivo. Our strategy was to reduce the expression of photoreceptor genes by targeting the rod-specific nuclear receptor Nr2e3 with a small molecule modulator. Treatment with PR3 decreased rod gene expression, and was sufficient to functionally and structurally preserve photoreceptors in the $Rho^{P23H}$ mouse. Previous studies have shown that genetic manipulation of the rod photoreceptor differentiation pathway may be useful for the treatment of multiple RP models. Conditional deletion of *Nrl* in adult mouse rods prevents degeneration in the $Rho^{-/-}$ model of recessive RP (*Montana et al., 2013*). More recently, knockdown of *Nrl* or *Nr2e3* by AAV-CRISPR/Cas9 gave long-term histological and functional preservation of photoreceptors in numerous RP models (*Zhu et al., 2017*; *Yu et al., 2017*). Our report now shows that a small molecule targeting this same regulatory pathway is also effective at slowing rod degeneration in a particularly aggressive RP model and provides a novel target for medical therapy of retinal degeneration.

# Materials and methods

### Key resource table

| Reagent type (species) or resource | Designation | Source or reference | Identifiers | Additional information |
|---|---|---|---|---|
| Transgenic mice (mus musculus), both sexes | *Rhodopsin-Pro23His* | The Jackson Laboratory | Jackson Stock No: 017628; RRID:IMSR_JAX:017628 | |
| Transgenic mice (mus musculus), both sexes | *Nr2e3-Rd7* | The Jackson Laboratory | Jackson Stock No: 004643; RRID:IMSR_JAX:004643 | |
| C57Bl6J mice (mus musculus), both sexes | Wild type | The Jackson Laboratory | Jackson Stock No: 000664; RRID:IMSR_JAX:000664 | |
| Chemical compound | Photoregulin3 (PR3) | ChemDiv, and then synthesized in Ding Lab | PubChem CID: 2092851 | |
| Chemical compound | Photoregulin1 (PR1) | ChemDiv, and then synthesized in Ding Lab | PubChem CID: 7901316 | |
| Antibody | Anti- Rhodopsin (Rho4D2), mouse monoclonal | Dr. Robert Molday, UBC | RRID:AB_2315273 | 1:250 |
| Antibody | Anti-S Opsin, goat polyclonal | SCBT | Sc-14363; RRID:AB_2158332 | 1:400 |

*Continued on next page*

*Continued*

| Reagent type (species) or resource | Designation | Source or reference | Identifiers | Additional information |
|---|---|---|---|---|
| Antibody | Anti Cone Arrestin, rabbit polyclonal | Millipore | AB15282; RRID:AB_1163387 | 1:1000 |
| Antibody | Anti-Otx2, goat polyclonal | R&D Systems | AF1979; RRID:AB_2157172 | 1:200 |
| Antibody | Anti-Nr2e3, mouse monoclonal | R&D Sysems: | PP-H7223-00; RRID:AB_2155481 | 1:10,000 |
| Antibody | Anti-Nrl, rabbit polyclonal | Chemicon | Ab5693 | 1:10,000 |
| Antibody | Anti-beta actin, mouse monoclonal | Abcam | Ab6276; RRID:AB_2223210 | 1:20,000 |
| Recombinant DNA | HsCD00084154 (Nr2e3) | DNASU | HsCD00084154 | |
| Recombinant DNA | BR-225Luc | Dr. Shiming Chen, Washington University; PMID: 15689355 | | |
| Recombinant DNA | S Opsin 600 pGL3 | Dr. Douglas Forrest, NIH; PMID: 16574740 | | |
| Recombinant DNA | pRL-CMV | Promega | E6931 | |
| Recombinant DNA | hNRL-pCMVSport6 | Open Biosystems | MHS1010-58005 | |
| Recombinant DNA | hCRX-pCMVSport6 | Open Biosystems | MHS1010-73672 | |
| Recombinant DNA | hNR2E3-pcDNA3.1/HisC | Dr. Shiming Chen, Washington University; PMID: 15689355 | | |
| Recombinant DNA | HsCD00329674 (RORB) | DNASU | HsCD00329674 | |

## Mice

C57Bl/6 (Jackson Stock No: 000664), $Rho^{P23H}$ (*Sakami et al., 2011*)(Jackson Stock No: 017628), and $Nr2e3^{Rd7}$ (Jackson Stock No: 004643) mice were used at the indicated ages. All mice were housed by the Department of Comparative Medicine at the University of Washington and protocols were approved by the University of Washington Institutional Animal Care and Use Committee. The research was carried out in accordance with the ARVO statement for the Use of Animals in Ophthalmic and Vision Research. For all experiments, a sample size of at least four mice per condition was chosen to ensure adequate power to detect a pre-specified effect size. From each litter, half of the animals were randomly assigned to the control group and the other half to the experimental group, and no animals were excluded.

## Photoregulin3

Photoregulin3 was identified by searching previous small molecule screens with PubChem for Nr2e3 interacting molecules. It was initially obtained from ChemDiv and then synthesized and purified in large quantities in the lab after initial screening. For in vivo experiments, mice were injected intraperitoneally with PR3 dissolved in DMSO at 10 mg/kg.

## Dissociated retinal cultures

Retinas were dissected from postnatal day 5 (P5) mice and dissociated by treatment with 0.5% Trypsin diluted in calcium- and magnesium-free HBSS for 10 min at 37°C. Trypsin was inactivated by adding an equal volume of FBS and cells were pelleted by centrifugation at 4°C and resuspended in media (Neurobasal-A containing 1% FBS, 1% N2, 1% B27, 1% Pen/Strep, and 0.5% L-Glutamine). For the immunofluorescence assay, cells were plated into 96-well black walled, clear bottom tissue culture plates at a density of 1 retina/5 wells. Small molecules were diluted in media and were added the day following dissociation. After two days of treatment, cells were fixed with 4% PFA for 20 min at room temperature, blocked with blocking solution (10% Normal Horse Serum and 0.5% Triton X-100 diluted in 1X PBS) for 1 hr at room temperature, and incubated overnight at 4°C with primary antibodies generated against Rhodopsin (1:250; Rho4D2, Gift from Dr. Robert Molday, UBC) diluted in blocking solution. The following day, wells were washed with 1X PBS and then incubated with species appropriate, fluorescently labeled secondary antibodies diluted in blocking solution for 1 hr at room temperature. Wells were washed three times, counterstained with ToPro3, and the entire plate was imaged using a GE Typhoon FLA 9400 imager. Optical density measurements were obtained

from the plate scans using ImageJ software and Rhodopsin expression was normalized to ToPro3 nuclear stain.

## Quantitative real-time PCR

RNA from retinas was isolated using TRIzol (Invitrogen) and cDNA was synthesized using the iScript cDNA synthesis kit (Bio-Rad). SSO Fast (Bio-Rad) was used for quantitative real-time PCR. For analysis, values were normalized to *Gapdh* ($\Delta C_T$) and $\Delta\Delta C_T$ between DMSO and compound-treated samples was expressed as percent of DMSO treated controls ($100*2\Delta\Delta Ct$). *t*-tests were performed on $\Delta C_T$ values. The following primer sequences were used: *Gapdh* (F: GGCATTGCTCTCAATGACAA, R: CTTGCTCAGTGTCCTTGCTG), *Rhodopsin* (F: CCCTTCTCCAACGTCACAGG, R: TGAGGAAGTTGA TGGGGAAGC), *Opn1sw* (F: CAGCATCCGCTTCAACTCCAA, R: GCAGATGAGGGAAAGAGGAA TGA), *Recoverin* (F: ACGACGTAGACGGCAATGG, R: CCGCTTTTCTGGGGTGTTTT), *Otx2* (F: CCGCCTTACGCAGTCAATG, R: GAGGGATGCAGCAAGTCCATA), and *Prdm1* (F: TTCTC TTGGAAAAACGTGTGGG, R: GGAGCCGGAGCTAGACTTG).

## Retinal explant cultures

Intact retinas without RPE from mice at the indicated ages and genotypes were explanted on 0.4 μm pore tissue culture inserts in media (Neurobasal-A containing 1% FBS, 1% N2, 1% B27, 1% Pen/ Strep, and 0.5% L-Glutamine) containing DMSO or 0.3–1.0 μM PR3. Full media changes were performed every other day.

## Whole mount staining

Explants or eye cups were fixed with 4% PFA for 20 min at room temperature, blocked with blocking solution (10% normal horse serum and 0.5% Triton X-100 diluted in 1X PBS) for 1 hr at room temperature, and incubated overnight at 4°C with primary antibodies generated against S Opsin (1:400, SCBT, sc-14363). The following day, the retinas were washed with 1X PBS, and then incubated with a species appropriate, fluorescently-labeled secondary antibody diluted in blocking solution overnight, followed by washing with 1X PBS and DAPI staining. The retinas were transferred to slides and coverslipped with Fluoromount-G (SouthernBiotech). An Olympus FluoView FV1000 was used for confocal microscopy. Cells were counted from single plane confocal images taken at fixed settings.

## Immunofluorescence

Eyecups were fixed in 4% PFA in 1X PBS for 20 min at room temperature and then cryoprotected in 30% sucrose in 1X PBS overnight at 4°C. Samples were embedded in OCT (Sakura Finetek), frozen on dry ice, and then sectioned at 16–18 μm on a cryostat (Leica). Slides were blocked with a solution containing 10% normal horse serum and 0.5% Triton X-100 in 1X PBS for 1 hr at room temperature and then stained overnight at 4°C with primary antibodies (Rho4D2 at 1:250 from Dr. Robert Molday, S Opsin at 1:400 from SCBT: sc-14363, Cone Arrestin at 1:1000 from Millipore: AB15282, Otx2 at 1:200 from R and D Systems: BAF1979) diluted in blocking solution. Slides were washed three times with 1X PBS the following day and then incubated in fluorescently labeled secondary antibodies diluted in blocking solution for 2 hr at room temperature, stained with DAPI, washed, and coverslipped using Fluoromount-G (SouthernBiotech). An Olympus FluoView FV1000 was used for confocal microscopy. Cells were counted from single plane confocal images taken at fixed settings. Counts in the central retina were taken adjacent to the optic nerve head (50 μm from the nerve head on the ventral side) and counts in the peripheral retina were taken 50 μm from the peripheral edge on the ventral side.

## Isothermal titration calorimetry

NR2E3 protein (aa 90–410) was expressed as an His8-MBP-TEV fusion protein from the expression vector pVP16 (DNASU Plasmid ID: HsCD00084154). *E. coli* BL21 (DE3) cells were grown to an OD600 of 1, and then induced with 0.2 mM IPTG at 16°C overnight. Cells were harvested, resuspended in extract buffer (20 mM Tris pH 8, 200 mM NaCl, 10% glycerol, 5 mM 2-mercaptoethanol, and saturated PMSF diluted 1:1,000), and then lysed by sonication on ice. Lysates were centrifuged at 4°C and the supernatant was loaded onto an equilibrated column containing 5 mL of Ni-NTA

agarose (Qiagen). The column was washed with 20 mM Tris pH 8, 1M NaCl, 5 mM 2-mercaptoethanol, and 40 mM imidazole, and then the protein was eluted with 20 mM Tris pH 8, 200 mM NaCl, 5 mM 2-mercaptoethanol, and 100 mM imidazole. The fusion protein was incubated with TEV overnight at 4°C and then the His8-MBP tags were separated from NR2E3 by ion exchange chromatography. For isothermal titration calorimetry, 100 µM PR3 was injected into 20 µM NR2E3 in 10 mM Sodium Phosphate buffer pH 8 with 50 mM NaCl and 0.5% DMSO in a MicroCal ITC-200 (Malvern) and the data was analyzed with Origin 7.0 software.

## RNA sequencing

RNA from retinas was isolated using TRIzol (Invitrogen) and total RNA integrity was checked using an Agilent 4200 TapeStation and quantified with a Trinean DropSense96 spectrophotometer. RNA-seq libraries were prepared from total RNA using the TruSeq RNA Sample Prep kit (Illumina) and a Sciclone NGSx Workstation (PerkinElmer). Library size distributions were validated using an Agilent 4200 TapeStation. Additional Library quality control, blending of pooled indexed libraries, and cluster optimization were performed using Life Technologies' Invitrogen Qubit Fluorometer. RNA-seq libraries were pooled (4-plex) and clustered onto a flow cell lane. Sequencing was performed using an Illumina HiSeq 2500 in rapid mode employing a paired-end, 50 base read length (PE50) sequencing strategy.

## Electron microscopy

Mice were euthanized by $CO_2$, and then perfused with 0.9% saline followed by 4% glutaraldehyde in 0.1 M sodium cacodylate buffer. Eye cups were fixed in 4% glutaraldehyde in 0.1 M sodium cacodylate buffer, washed with 0.1 M sodium cacodylate buffer, and then post-fixed in 2% osmium tetroxide. After fixation, eye cups were washed with water, dehydrated through a graded series of ethanol, incubated in propylene oxide and then epon araidite, polymerized overnight at 60°C, and then sectioned at a thickness of 70 nm. Images were obtained using a JEOL JEM-1230 electron microscope.

## ERGs

Mice were dark adapted overnight (12–18 hr). All subsequent steps were carried out under dim red light. Mice were placed in an anesthesia chamber and anesthetized with 1.5–3% isoflurane gas. Mice were transferred from the anesthesia chamber to a heated platform maintained at 37 °C and positioned in a nose cone to maintain a constant flow of Isoflurane. Drops of 1% tropicamide and 2.5% phenylephrine Hydrochloride were applied to each eye. A reference needle electrode was placed subdermally on the top of the head and a ground needle electrode was placed subdermally in the tail. Drops of 1.5% methyl cellulose were applied to each eye and contact lens electrodes were placed over each eye.

Dim red light was turned off and the platform was positioned inside of an LKC Technologies UTAS BigShot ganzfeld and a series of flashes of increasing intensity were delivered scotopically. A series of photopic flashes were performed immediately following the series of scotopic flashes.

## Dual luciferase assay

HEK293 cells were transfected with 1 µg of the luciferase reporter BR-225Luc (Dr. Shiming Chen) or S Opsin 600 pGL3 (Dr. Douglas Forrest), 1 ng of the control pRL-CMV (Promega) and 100 ng of hNRL-pCMVSport6 (Open Biosystems), hCRX-pCMVSport6 (Open Biosystems), hNR2E3-pcDNA3.1/HisC (Dr. Shiming Chen), or hRORB-pLenti6.2/V5-DEST (DNASU) in 24-well plates using Lipofectamine 3000 reagent (Thermo Fisher Scientific). Transfection reagents were removed the following day and replaced with media containing DMSO or 1 µM PR3 for 2 days. Cells were lysed and firefly and renilla luciferase activity was measured with the Dual-Luciferase Reporter Assay System (Promega) using a 1420 Multilabel Victor3V plate reader.

## Co-immunoprecipitation

HEK293 cells were transfected in 6-well tissue culture plates with lipofectamine 3000 (Thermo Fisher Scientific) and 800 ng of each hNRL-pCMVSport6 (Open Biosystems) and hNR2E3-pcDNA3.1/HisC (Dr. Shiming Chen) in Opti-MEM media. Transfection reagents were removed after 24 hr and

replaced with media containing DMSO or 1 µM PR3 for 2 days. Cells were lysed with Co-IP lysis buffer (25 mM Tris-HCl pH 7.5, 150 mM NaCl, 1 mM EDTA, 1% Triton X-100, 5% glycerol and 1X protease inhibitor cocktail). Sheep anti-mouse IgG magnetic Dynabeads were incubated with anti-Nr2e3 antibody (5 µg/precipitation) diluted in Co-IP buffer for 2 hr at 4°C. Equal volume of lysate were then added to the antibody-coated beads and incubated overnight at 4°C. The following day, beads were washed four times with Co-IP buffer and then incubated at 85°C for 15 min in 1X sample buffer diluted in Co-IP buffer. Samples were loaded and run in a 4–20% SDS gel (Bio-Rad). Protein was transferred to a PVDF membrane (Thermo Fisher Scientific), blocked (5% BSA and 0.1% Tween 20 in 1X PBS) for at least 1 hr at room temperature and stained with primary antibodies anti-Nrl (Chemicon) or Anti-Nr2e3 (R and D systems) diluted in blocking solution overnight at 4°C. Membranes were washed with 0.1% Tween 20 in 1X PBS and then incubated with Clean-Blot IP Detection Reagent (Thermo Fisher Scientific) diluted in blocking solution for 1 hr at room temperature. Signals were visualized on X-ray film with SuperSignal West Dura Extended Duration Substrate (Thermo Fisher Scientific) and quantified using ImageJ software.

## Acknowledgements

The authors acknowledge the support of NIH R01 EY021374 (to T Reh, S Ding and K Zhang), 1PO1 GM081619-01 (T Reh) and the Allen Distinguished Investigator Award (T Reh). The authors thank members of the Reh and Berminghan-McDonogh labs for helpful critique, and Dr. John Sumida and members of Drs. Jason Smith and Wenqing Xu labs for technical assistance and advice, and Dr. Timothy Cherry for comments on the manuscript. The authors also thank Jonathan Linton from the Vision Core for assistance with ERGs.

## Additional information

### Funding

| Funder | Grant reference number | Author |
|---|---|---|
| National Institute of General Medical Sciences | | Thomas A Reh<br>Paul A Nakamura<br>Andy A Shimchuk<br>Kole DeGolier |
| Paul G. Allen Family Foundation | | Thomas A Reh<br>Paul A Nakamura<br>Andy A Shimchuk<br>Kole DeGolier |
| National Eye Institute | | Thomas A Reh<br>Paul A Nakamura<br>Andy A Shimchuk<br>Kole DeGolier |
| National Eye Institute | EY021374 | Thomas A Reh<br>Paul A Nakamura<br>Shibing Tang<br>Zhizhi Wang |

The funders had no role in study design, data collection and interpretation, or the decision to submit the work for publication.

### Author contributions

Paul A Nakamura, Conceptualization, Resources, Formal analysis, Supervision, Funding acquisition, Project administration, Writing—review and editing; Andy A Shimchuk, Conceptualization, Resources, Data curation, Formal analysis, Supervision, Validation, Investigation, Visualization, Methodology, Writing—original draft, Project administration; Shibing Tang, Sheng Ding, Formal analysis, Investigation, Methodology; Zhizhi Wang, Conceptualization, Resources, Methodology; Kole DeGolier, Resources, Validation, Methodology; Thomas A Reh, Conceptualization, Resources, Formal analysis, Supervision, Funding acquisition, Project administration

## Author ORCIDs

Paul A Nakamura ⃝iD http://orcid.org/0000-0002-3845-8477
Thomas A Reh ⃝iD http://orcid.org/0000-0002-3524-0886

## Ethics

Animal experimentation: All experiments were performed in strict accordance with the protocols approved by the University of Washington Institutional animal care and use committee (IACUC).

## Decision letter and Author response

Decision letter https://doi.org/10.7554/eLife.30577.014
Author response https://doi.org/10.7554/eLife.30577.015

## Additional files

### Supplementary files

• Transparent reporting form
DOI: https://doi.org/10.7554/eLife.30577.013

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
