## [Decision Letter]

Thank you for submitting your manuscript "Small molecule Photoregulin3 prevents retinal degeneration in the *Rho^P23H^* mouse model of retinitis pigmentosa" to *eLife*. The manuscript has been reviewed by three expert reviewers, and their assessments together with my own (Jeremy Nathans) as Reviewing Editor, forms the basis of this letter. The process was overseen by Senior editor Gary Westbrook. I am also including the three reviews in their original form at the end of this letter, as there are many specific and useful suggestions in them that will not be repeated in the summary here. The following individuals involved in review of your submission have agreed to reveal their identity: Joseph C Corbo (Reviewer #2) and Bo Chen (Reviewer #3).

The reviewers and I think that the identification of small molecule Nr2e3 antagonists is important and that exploring the application of these antagonists in a rod-to-cone conversion strategy for treatment of RP is exciting. However, we have concerns about the incompleteness of the data and its interpretation. In particular, our central concern is that it is not clear whether PR3 action in vivo is through Nr2e3 or might be an off-target effect (or a combination of the two).

Reproduced below is an edited version of our on-line discussion that captures this critique and includes a specific proposal for a follow-up experiment with the Nr2e3 KO (=rd7) mouse that would clarify the target of PR3. We think that this experiment would greatly enhance the value of your manuscript. [A more extensive experiment in which the rd7 allele is crossed into the P23H background would also be enlightening but would take substantially longer, assuming that one is starting from scratch with the crosses.]

"There is an apparent discrepancy between the Nr2e3 KO phenotypes and the implied mechanisms of PR3 rescuing P23H degeneration, namely the mild rod gene reduction and strong cone gene derepression in the Nr2e3 KO versus the strong rod gene reduction and relatively normal cone gene expression in the in vivo PR3 treatment. A discrepancy between a genetic KO and pharmacological perturbation certainly could be due to interesting real biology (elaborated below), but could also be a red flag for the not-so-uncommon off-target effects. In either case, I feel it is necessary to establish whether PR3 acts through NR2E3. This is important because if off-target, it requires a major reinterpretation and because in its current form, the manuscript seems misleading for the field by equating Nr2e3 inhibition to Nrl inhibition with regard to retarding retinal degeneration.

As pointed out already, we could ask for data showing whether genetic deletion of Nr2e3 is protective in the P23H model. If the data are unavailable or contradictory (i.e. no protection), the observed rescue could be due to effects from acute pharmacological inhibition that can only be modeled by a conditional knock-out model (e.g. some compensatory mechanism leading to a milder phenotype in the germline Nr2e3 null). Alternatively, PR3 might function as an inverse agonist and induce some gain of function from the NR2E3 protein, as Joe suggested. There are at least two conceivable scenarios. First, NR2E3 binds to both rod and cone gene promoters and functions as a weak activator for rod genes (possibly inhibited by NRL/CRX) but a strong repressor for cone genes (on its own). PR3 somehow locks NR2E3 into a strong repressor on both rod and cone promoters. Second, PR3 enhances interaction between NR2E3 and NRL/CRX such that PR3-bound NR2E3 depletes NRL/CRX that would otherwise activate rod genes. Interestingly, their first compound PR1 was reported in their IOVS paper to enhance NR2E3/NRL/CRX complex formation in a heterologous system, although no explanation was given. Regardless of the actual mechanisms, the effect of PR3 in reducing rod genes beyond the Nr2e3 genetic null should depend on the presence of NR2E3 protein; this could be tested by treating Nr2e3 KO with PR3. Specifically, one could measure the reduction in rhodopsin expression under the treatment regimen (not just 24 hrs) in wildtype mice; this reduction is expected to disappear in the Nr2e3 KO if PR3 acts through NR2E3. This doesn't require crossing mice and should be do-able within a reasonable time frame. This could at least rule out a trivial explanation of the rescue: non-specific developmental delay in mutant rhodopsin expression and hence delay in manifestation of retinal degeneration. In any event, the possibility of off-target effects should certainly be discussed, and the 67 μm Kd in ITC explained."

Reviewer #1:

This manuscript is an addition to the emerging theme of treating rod degeneration via rod-to-cone reprogramming. Proof-of-principle studies by other labs showed that genetic deletion of Nrl leads to functional improvement in multiple mouse models of retinal degeneration. The authors sought to pharmacologically inhibit Nr2e3, a transcription factor downstream of Nrl, and identified in a chemical screen several Nr2e3 inhibitors, one of which (PR1) they have published and inhibits rod gene expression while increasing cone gene expression in culture. In this manuscript, the authors reported a structurally unrelated chemical (PR3) and tested its effect on rod/cone gene expression and rescuing retinal degeneration in a mouse model carrying a dominant rhodopsin mutation. While the clinical significance and striking rescue phenotypes would be appropriate for *eLife*, this reviewer would like to see additional evidence to rule out off-targets given the potential discrepancy between genetic and pharmacological inactivation of Nr2e3.

The Nrl mutant retinas expresses little rhodopsin, but the Nr2e3 mutant retinas have a very limited decrease in rhodopsin expression, especially at 2-3 postnatal weeks that are of interest here. This, together with the likely partial inhibition characteristic of pharmacological approaches, raises the question whether a partial inhibition of Nr2e3 would lead to a functionally-relevant decrease in mutant rhodopsin expression, as achieved via genetic deletion of Nrl. In the absence of genetic deletion of Nr2e3 (similar to the Nrl experiments), a necessary experiment would be to measure rhodopsin downregulation in wild type mice under the same treatment regimen as in the rhodopsin mutant mice (not just 24 hr post treatment at a different postnatal day). How do the authors rule out the possibility that the rescue of degeneration is simply due to a non-specific delay in development and hence a delay in degeneration? Other retinal cells might not be the best control if they are not actively developing during this time window. Are there other organs/cells known to develop rapidly at the time of PR3 treatment?

Related to this, this reviewer is confused about why the ITC Kd measuring the direct interaction is 100 times higher than the effective biological concentration in culture.

This reviewer doesn't see "rod nuclei of PR3-treated retinas contained smaller patches of densely packed heterochromatin" in Figure 2. This needs to be quantified or deleted.

The original chemical screen is based on disruption of the interaction between Nr2e3 and a transcriptional co-repressor, NCoR. It should be discussed how such disruption and presumable loss of transcriptional inhibition lead to, counterintuitively, a decrease in rhodopsin expression and/or disruption of the suggested Nrl/Nr2e3/Crx complex.

Reviewer #2:

In this exciting study, the authors present the identification of a small molecule (photoregulin3 or PR3) which is capable of retarding the progression of photoreceptor degeneration in the RhodopsinP23H mutant mouse, a model for retinitis pigmentosa. This study is a follow-up on a previous study from the same group (Nakamura et al., 2016) which identified another small molecule, photoregulin1, with related properties. These two reports form part of a growing body of work from several groups which suggests that genetic or chemical inhibition of the transcriptional pathway regulating rod gene expression (Nrl -> Nr2e3) can confer protection against photoreceptor degeneration in a range of genetic models of blindness. The present study is important because it is the first to demonstrate efficacy of a small molecule (PR3), putatively targeting this pathway, in preventing photoreceptor degeneration in vivo.

There are several issues that, if addressed, would strengthen the conclusions of the study. First, some of the experiments appear to have been performed in very few, or even a single, biological replicate. For example, the data presented in Figure 1 (the effects of PR1 and PR3 on Rho expression by qPCR) show a single biological replicate. To solidify this result, three biological replicates should be presented. Similarly, only two biological replicates are presented in Figure 1. I suspect that addition of a third biological replicate is unlikely to materially affect the conclusions derived from this experiment, but I would normally have expected at least three biologic replicates for an experiment of this sort. It was unclear from the text how many biological replicates were performed for the RNA-seq experiment presented in Figure 2. At a minimum, two biological replicates are needed and three are preferred.

Reviewer #3:

Nakamura et al. presented results reporting that photoregulin3 (PR3), identified from a small molecule screen as an inhibitor of Nr2e3, regulates photoreceptor gene expression and morphology in the wildtype retinas, and very interestingly, in vivo treatment with PR3 preserves both rod and cone photoreceptors and rescues retinal function in Rho-P23H mice, a knock-in mouse model of autosomal dominant retinitis pigmentosa (adRP). Ne2e3 is located downstream of Nrl to promote rod cell fate while suppressing the expression of cone genes. Reduction of Nrl expression by targeted deletion (Matana et al., 2013) or Crispr/Cas9-mediated knockdown (Yu et al., 2017) retarded retinal degeneration and partially preserved retinal function, through a mechanism that rods could be partially converted to cones in the adult mouse retina. The authors presented convincing evidence that similar to Nrl knockdown, chemical inhibition of Ne2e3 by PR3 suppressed rod phenotype, although not completely clear if rods were undergoing conversion to cones, and prevented photoreceptor death in Rho-P23H mice at young age (postnatal day 21). The research is well presented with a potentially high impact for the treatment of adRP with PR3.

Here are some points to consider and to make sure the interpretations are thoroughly supported:

1) The P23H homozygous mice were used in the retinal degeneration studies. The use of P23H/WT rhodopsin heterozygous mice would have been more appropriate to mimic the human disease as the P23H mutated protein is co-expressed with the wild-type rhodopsin protein.

2) Please show representative images that PR3 treatment reduced rhodopsin expression in dissociated retinal cultures.

3) In Figure 1, the authors showed that PR3 treatment enhanced s-opsin expression in the ventral retina. What happens to the s-opsin expression in the dorsal retina?

4) Does IP injection of PR3 in intact mice at P13 lead to upregulation of s-opsin as happens to the explants culture?

5) Shown in Figure 2, the authors claimed that PR3 treatment affected photoreceptor outer segment development. The outer segments consist of densely packed membrane discs where visual pigments and essential proteins for phototransduction are housed. Do membrane discs appropriately form in the PR3-treated retinas? Please show high-resolution EM pictures to clarify.

6) To rescue photoreceptors in P23H homozygous mice, PR3 treatment started at P12-P14, and the results were analyzed a week later at P21. adRP caused by P23H mutation is a chronic degenerative condition, commonly spanning decades in human patients. It is interesting to know how long the protective effects last. Are photoreceptors still protected when the treated mice are analyzed at 6 or 8 weeks of age?

---

## [Author Response]

[…] “Specifically, one could measure the reduction in rhodopsin expression under the treatment regimen (not just 24 hrs) in wildtype mice; this reduction is expected to disappear in the Nr2e3 KO if PR3 acts through NR2E3. This doesn't require crossing mice and should be do-able within a reasonable time frame. This could at least rule out a trivial explanation of the rescue: non-specific developmental delay in mutant rhodopsin expression and hence delay in manifestation of retinal degeneration. In any event, the possibility of off-target effects should certainly be discussed, and the 67 μm Kd in ITC explained."

Thank you for your thoughtful reviews and critiques. We have carried out most of the experiments you and the reviewers have suggested; many of these concerns had to do with potential other targets of PR3 and we have done our best to evaluate this possibility. Since we cannot rule out that there may be other targets in the retina for PR3, we have modified the Abstract and text in many places to leave open this alternate interpretation. Nevertheless, we were able to carry out the critical experiment that was raised in the Discussion section with the reviewers and the results support the hypothesis that Nr2e3 is at least one important target of PR3:

As requested by the reviewers we have carried out the above experiment. We were not able to obtain the Nr2e3 knockout mouse, because this strain is no longer available; however, the Rd7 mouse has a spontaneous insertion in the Nr2e3 gene and is considered a null in the field. We found that PR3 no longer has a significant effect on *Rhodopsin* expression in the Rd7 retina. The results are described in the new text added to the manuscript (Results and Discussion, fifth paragraph):

"To further evaluate whether PR3 has other targets in the retina, we tested PR3 in Rd7 retinas, which harbor a spontaneous mutation in Nr2e3 [27-29]. We explanted adult retinas from wild type and Rd7 mice in media containing DMSO or PR3 and measured Rhodopsin expression by qPCR. Consistent with the hypothesis that the effects of PR3 on rod gene expression are mediated through Nr2e3, PR3 caused a significant reduction in Rhodopsin expression in wild type retinas, but not in retinas from Rd7 mice (Figure 1). While this experiment confirms that Nr2e3 is a key target of PR3, it remains unclear why Rd7 retinas exhibit only moderate reductions in rod gene expression [11-13, 28], while PR3 treatment or CRISPR-Cas9 deletion of Nr2e3 [14] result in substantial decreases. This suggests that there is some developmental compensation that is not present when the gene is deleted or inhibited in postmitotic rods; alternatively, PR3 may be acting in a dominant-negative manner, and inhibiting the ability of the Nr2e3-Nrl-Crx complex to function properly."

Reviewer #1:[…] The Nrl mutant retinas expresses little rhodopsin, but the Nr2e3 mutant retinas have a very limited decrease in rhodopsin expression, especially at 2-3 postnatal weeks that are of interest here. This, together with the likely partial inhibition characteristic of pharmacological approaches, raises the question whether a partial inhibition of Nr2e3 would lead to a functionally-relevant decrease in mutant rhodopsin expression, as achieved via genetic deletion of Nrl. In the absence of genetic deletion of Nr2e3 (similar to the Nrl experiments), a necessary experiment would be to measure rhodopsin downregulation in wild type mice under the same treatment regimen as in the rhodopsin mutant mice (not just 24 hr post treatment at a different postnatal day). How do the authors rule out the possibility that the rescue of degeneration is simply due to a non-specific delay in development and hence a delay in degeneration? Other retinal cells might not be the best control if they are not actively developing during this time window. Are there other organs/cells known to develop rapidly at the time of PR3 treatment?

We also find it interesting that Photoregulin3 (PR3) treatment leads to such a large change in rod gene expression, whereas the germline deletion of *Nr2e3* or the Rd7 mutation does not; however, this may be due to compensation from loss of the gene in development by other rod transcription factors, and this compensation may not occur in PR3 treated retinas or adult deletion of *Nr2e3*.

Since we submitted our manuscript, we became aware of another study that used CRISPR-Cas9 to delete *Nr2e3* in postnatal mice (Zhu et al., 2017); deletion of this gene in adult photoreceptors caused a large reduction in rod gene expression and an increase in cone gene expression, very similar to the CRISPR-Cas9 deletion of *Nrl* (Figure S1 of Zhu et al., 2017).

Furthermore, this recent study demonstrated that deletion of Nr2e3 by CRISPR-Cas9 is protective in the Rd1 and Rd10 models of RP (Figure 1 of Zhu et al., 2017), similar to what was found with *Nrl* deletion and similar to the effects we observe with PR3 on *Rho^P23H^* mice. Thus, the CRISPR-Cas9 deletion of *Nr2e3* shows that targeting Nr2e3 can cause a functionally relevant decrease in rod gene expression.

As for whether PR3 causes a non-specific delay in development of the animal or the retina, we have not observed any change in body mass or age of eye opening in treated mice, and the other cells in the retina appear to be similar in both treated and control mice. Additionally, the RNAseq shows primarily changes in photoreceptor genes. For example, we see no change in the expression of Muller glial genes *Rlbp1, Slc1a3* (Glast), or *Glul*. Muller glia are the last-born cells of the retina and are still maturing at this time.

To further control for this possibility we tested PR3 on adult wild type retinas in explant cultures. We found that after two days there was a 60% reduction in *Rhodopsin* expression in the PR3 treated retinas, as compared to the DMSO control. These data are now shown in Figure 1.

Related to this, this reviewer is confused about why the ITC Kd measuring the direct interaction is 100 times higher than the effective biological concentration in culture.

For ITC, we used Nr2e3 protein purified from *E. coli*. Proteins purified from bacterial cultures do not have any post-translation modifications and may not be completely folded correctly, and these differences may contribute to this discrepancy. It is also possible that Nr2e3 changes conformation in the presence of co-activators or co-repressors and PR3 then has a higher affinity. In HEK 293 cells, transfected with Nr2e3, Crx and Nrl, we see effects of PR3 at 1 μM (Figure 1—figure supplement 1). Additionally, the original screen used a different mammalian cell line (CHO-S) transfected with Nr2e3 and NCoR and calculated an IC_50_ of 0.07 μM for this compound (PubChem Assay ID 624394).

This reviewer doesn't see "rod nuclei of PR3-treated retinas contained smaller patches of densely packed heterochromatin" in Figure 2. This needs to be quantified or deleted.

We agree this is not obvious from the micrographs shown in the previous version. We see some differences in the treated retinas in regards to nuclear shape and ultrastructure, but they are not easily quantifiable, and so we have deleted this statement.

The original chemical screen is based on disruption of the interaction between Nr2e3 and a transcriptional co-repressor, NCoR. It should be discussed how such disruption and presumable loss of transcriptional inhibition lead to, counterintuitively, a decrease in rhodopsin expression and/or disruption of the suggested Nrl/Nr2e3/Crx complex.

The reviewer is correct that the results we observe on rod gene expression would not necessarily be predicted from a screen that assayed the interactions between Nr2e3 and a co-repressor. We have therefore added the following text to the manuscript (Results and Discussion, fourth paragraph):

"The initial screen that identified PR3 demonstrated its effects on Nr2e3 in a co-repression assay with NCoR; however, the effects we observed on rod gene expression suggested that PR3 also inhibits the co-activator function of Nr2e3. […] It is not clear why stabilizing the complex would reduce its activity; however, it may be that stabilizing interactions with Nr2e3 prevents Nrl and Crx from interacting with consensus sites on the DNA, or alternatively prevents the recruitment of other components of the transcriptional machinery."

Reviewer #2:There are several issues that, if addressed, would strengthen the conclusions of the study. First, some of the experiments appear to have been performed in very few, or even a single, biological replicate.For example, the data presented in Figure 1 (the effects of PR1 and PR3 on Rho expression by qPCR) show a single biological replicate. To solidify this result, three biological replicates should be presented. Similarly, only two biological replicates are presented in Figure 1.

The reviewer is correct that the data shown in Figure 1 were from a single biological replicate and the data from Figure 1 was done in duplicate. Figure 1 is now shown as an average of 3 biological replicates, and we performed the qPCR analysis at a single dose with 3-4 biological replicates per condition and is now shown in Figure 1.

It was unclear from the text how many biological replicates were performed for the RNA-seq experiment presented in Figure 2. At a minimum, two biological replicates are needed and three are preferred.

We apologize for any confusion. The RNAseq data shown is an average of two biological replicates, and we have stated this more clearly in the text and the legend.

Reviewer #3:1) The P23H homozygous mice were used in the retinal degeneration studies. The use of P23H/WT rhodopsin heterozygous mice would have been more appropriate to mimic the human disease as the P23H mutated protein is co-expressed with the wild-type rhodopsin protein.

We agree that using heterozygous mice might be more appropriate to mimic the human disease. However, the *Rho^P23H^* heterozygous mice do not show significant degeneration until postnatal day 148 (Sakami et al., 2011), which is a bit prohibitive because we need to inject PR3 daily to repress rod gene expression. In addition, we find that the pharmacokinetic properties of the compound are such that we cannot reach an effective dose in older mice by IP injections, and are limited in the number of intravitreal injections possible in mice. Nonetheless, we see our rescue in the homozygous mice as an in vivo proof-of-principle for this approach, and we are currently working on optimizing our compounds for oral bioavailability to test in the heterozygotes.

2) Please show representative images that PR3 treatment reduced rhodopsin expression in dissociated retinal cultures.

We have added these images to Figure 1.

3) In Figure 1, the authors showed that PR3 treatment enhanced s-opsin expression in the ventral retina. What happens to the s-opsin expression in the dorsal retina?

We do not see a change in the dorsal retinal (now shown in Figure 1).

4) Does IP injection of PR3 in intact mice at P13 lead to upregulation of s-opsin as happens to the explants culture?

We see a similar effect after IP injection of PR3 in wild type mice (now shown in Figure 2), an increase in S Opsin+ cells in the ventral but not dorsal retina.

5) Shown in Figure 2, the authors claimed that PR3 treatment affected photoreceptor outer segment development. The outer segments consist of densely packed membrane discs where visual pigments and essential proteins for phototransduction are housed. Do membrane discs appropriately form in the PR3-treated retinas? Please show high-resolution EM pictures to clarify.

The PR3 treated retinas do not have outer segments, and this is now shown more clearly in Figure 2.

6) To rescue photoreceptors in P23H homozygous mice, PR3 treatment started at P12-P14, and the results were analyzed a week later at P21. adRP caused by P23H mutation is a chronic degenerative condition, commonly spanning decades in human patients. It is interesting to know how long the protective effects last. Are photoreceptors still protected when the treated mice are analyzed at 6 or 8 weeks of age?

With the current compound, PR3, we are unable to treat for 6-8 weeks due to poor PK (animals need to be dosed daily). However, Crispr-Cas9 deletion of either *Nrl* (Yu et al., 2017) or *Nr2e3* (Zhu et al., 2017) provides protection for many weeks in several RP models. We are currently working on modifying our compounds for better bioavailability to test for long-term protection.